# Learning by Exclusion: An Evidential Contrapositive Framework for Zero-Shot OOD Detection

## Abstract

Out-of-Distribution (OOD) detection is critical for deploying vision-language models in safety-sensitive settings. While recent approaches such as CLIPN rely solely on margin-based cosine similarity for separating in- and out-of-distribution samples, this reliance provides limited safeguards against overconfident representations. We introduce the Evidential Contrapositive Framework (ECF), a principled approach that integrates logical negation into vision-language alignment to explicitly model what a class is not. Unlike prior work that enforces separation through heuristic margins, ECF specifically introduces (i) exclusion via contrapositive loss, (ii) decoupling via negative prompt alignment, and (iii) logical consistency via contrapositive similarity regularization. To further enhance reliability, we couple this framework with evidential uncertainty modeling using a Dirichlet distribution, enabling simultaneous estimation of aleatoric and epistemic uncertainty. This combination yields interpretable uncertainty-aware decision boundaries and robust rejection of OOD inputs without requiring access to OOD samples during training. Extensive experiments on large-scale benchmarks demonstrate that ECF significantly outperforms state-of-the-art zero-shot OOD methods, both in detection accuracy and in uncertainty calibration, validating the advantage of principled contrapositive reasoning over margin-based objectives.

## 1 Introduction

The ability to distinguish between in-distribution (ID) and out-of-distribution (OOD) inputs is critical for deploying machine learning models in safety-sensitive applications such as autonomous driving (Huang et al., 2020), medical diagnostics (Hong et al., 2024), and biodiversity monitoring (Impiö & Raitoharju, 2024). Traditional OOD detection methods assume access to labeled training data from known classes, but this assumption breaks down in zero-shot settings, where the model must generalize to unseen classes without direct supervision. Recent advancements in vision-language models (VLMs), particularly Contrastive Language–Image Pre-trained (CLIP) (Radford et al., 2021), have demonstrated strong zero-shot recognition capabilities, prompting their use in OOD detection tasks. Despite these advances, current zero-shot OOD detection techniques, including recent works like CLIPN (Wang et al., 2023), struggle with reliable rejection mechanisms. They often rely on thresholding similarity scores without explicitly modeling uncertainty or accounting for logical relationships between classes and prompts. This leads to overconfident predictions on OOD inputs, compromising both reliability and safety.

We propose a novel framework called Learning by Exclusion: Evidential Contrapositive Framework (ECF) to address these limitations. Inspired by contrapositive reasoning in logic: *where the truth of a statement is inferred through the falsity of its inverse*, we design a learning objective that encourages the model to learn class-representative embeddings by explicitly pushing away embeddings from non-class (negative) prompts. Additionally, we incorporate an evidential uncertainty modeling mechanism based on Dirichlet distributions, enabling the model to express both epistemic and aleatoric uncertainty. This results in a principled rejection-aware training pipeline.

Specifically, ECF introduces three core components: (1) *Exclusion-Aware Prompt Engineering*, which generates negative prompts to enforce semantic dissimilarity from non-target classes; (2)

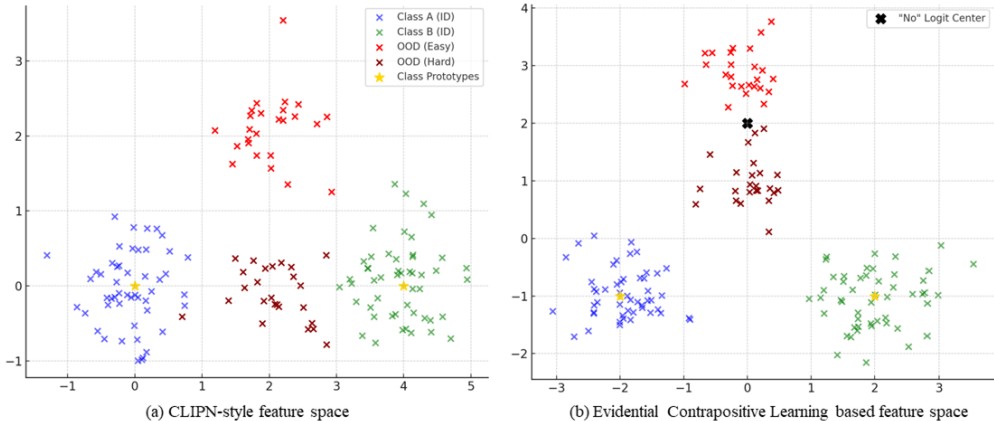

Figure 1: Toy t-SNE visualization comparing the feature space of (a) CLIPN (Wang et al., 2023) and (b) our proposed ECF method. (a) CLIPN-style OOD Detection: OOD samples (especially hard ones) are often close to in-domain classes; (b) ECF based OOD Detection: OOD samples are repelled toward a designated "No" region, improving separation; Hard-to-distinguish OOD samples overlap with ID regions in (a), while in (b), they are repelled toward a distinct "No" logit region.

*Negative Prompt Alignment Loss*, a novel objective that penalizes similarity between OOD features and ID class prototypes; and (3) *Contrapositive Similarity Regularization*, which ensures the model explicitly aligns dissimilar features with dissimilar textual prompts. These components are unified under a rejection-aware learning framework that supports high-confidence abstention on OOD samples. Fig 1 shows the feature spaces of CLIPN and our proposed evidential contrapositive Framework (ECF). While CLIPN exhibits partial separation between ID and OOD samples via prompt ensembling and threshold tuning, ECF achieves significantly better separation through contrapositive learning and evidential uncertainty modeling, highlighting its superior zero-shot OOD discrimination capability.

Our framework achieves state-of-the-art performance on zero-shot OOD detection across multiple benchmark datasets. It not only improves discriminative power but also enables interpretable uncertainty quantification, making it well-suited for real-world open-world scenarios. Our contributions are summarized as follows:

- We introduce Evidential Contrapositive Framework (ECF), a framework that enables zero-shot out-of-distribution (OOD) detection by leveraging evidential uncertainty modeling and contrapositive reasoning, eliminating the need for OOD samples during training.

- ECF incorporates evidential uncertainty estimation into vision-language models, providing calibrated confidence scores that enhance the model's ability to distinguish between in-distribution and out-of-distribution samples.

- We propose a contrapositive learning approach that utilizes logical negation to generate pseudo-negative samples, strengthening the model's discriminative capabilities without requiring additional data.

- Our extensive experiments demonstrate that ECF achieves superior performance in zero-shot OOD detection tasks, outperforming state-of-the-art methods.

## 2 RELATED WORK

In this section, we review existing literature relevant to our work across the following themes: (i) Zero-Shot Out-of-Distribution (OOD) Detection, (ii) Prompt-Based Vision-Language Models, (iii) Uncertainty-Aware Deep Learning, and (iv) Logical and Contrastive Reasoning in Representation Learning. We then contextualize our proposed approach by contrasting it with CLIPN (Wang et al., 2023).

## 2.1 ZERO-SHOT OOD DETECTION

Zero-shot OOD detection aims to identify anomalous or novel classes without requiring exposure to OOD examples during training. Earlier works such as ODIN (Liang et al., 2018), Mahalanobis distance-based scores (Lee et al., 2018), and energy-based models (Liu et al., 2020) predominantly focused on confidence manipulation or statistical modeling of in-distribution features.

Most of the zero-shot approaches leverage vision-language models like CLIP (Radford et al., 2021) to compare image embeddings with natural language descriptions of known classes. To improve CLIP's OOD capabilities, methods such as CLIPN (Wang et al., 2023) have been proposed. CLIPN fine-tunes CLIP using synthetic negatives and introduces rejection calibration via prompt ensembling and adaptive thresholds. Recent works have explored leveraging vision–language models (VLMs) for OOD detection Ming et al. (2022); Jiang et al. (2024); Zhang & Zhang (2024). For instance, MCM Ming et al. (2022) computes Mahalanobis distances in the CLIP embedding space and derives a handcrafted confidence score, but does not incorporate contrastive objectives or uncertainty modeling. NegLabel Jiang et al. (2024) employs negative prompts as pseudo-labels to improve discrimination, yet all negatives are treated uniformly and the framework lacks a principled mechanism for modeling epistemic or aleatoric uncertainty, which can limit robustness. AdaNeg Zhang & Zhang (2024) extends this idea by adaptively selecting negative prompts, but it still does not enforce explicit contrapositive alignment or disentangled evidence estimation. In contrast, our approach introduces an evidential contrapositive framework that combines logical reasoning with uncertainty-aware modeling, enabling structured exclusion, decoupling, and consistency that go beyond margin-based or purely negative-sampling strategies.

## 2.2 PROMPT-BASED VISION-LANGUAGE MODELS

Vision-language models, particularly those leveraging prompt-based mechanisms, have become central to modern representation learning due to their scalability and flexibility in zero-shot tasks. Models such as CLIP (Radford et al., 2021) train a dual encoder architecture to align textual prompts with visual features via contrastive learning. By transforming class labels into textual prompts (e.g., "a photo of a *cat*"), these models project both modalities into a shared embedding space. Zero-shot classification is then achieved by comparing the image embedding with the prompt embeddings of all candidate classes. Subsequent works have extended this paradigm to address OOD detection. CLIP-Adapt (Gudibande et al., 2023) and Tip-Adapter (Zhang et al., 2022) refine CLIP's zero-shot capabilities by adapting prompt embeddings or employing retrieval-based mechanisms. These methods, while improving classification, often lack mechanisms to reject unknown or spurious inputs, resulting in poor OOD detection performance.

Recent approaches such as CLIPN (Wang et al., 2023) introduce negative prompts and an auxiliary "None of the Above" class to enable rejection, offering one path toward OOD-aware vision-language systems. However, CLIPN relies on learned class centroids and post-hoc calibration, limiting generalizability to unseen OOD distributions. In contrast, our proposed Evidential Contrapositive Framework (ECF) integrates logical contrapositive reasoning with evidential uncertainty modeling to perform zero-shot OOD detection by learning representations that are explicitly aware of counterfactual prompts and uncertainty. This fundamental distinction allows ECF to generalize better across domains and classes without reliance on synthetic negatives or additional calibrations.

## 2.3 UNCERTAINTY-AWARE DEEP LEARNING

Uncertainty estimation is crucial for detecting OOD inputs. It is broadly categorized into two types: *epistemic uncertainty*, which captures model uncertainty due to limited data, and *aleatoric uncertainty*, which accounts for noise inherent in the data (Kendall & Gal, 2017).

Bayesian Neural Networks (BNNs) (Blundell et al., 2015) and Monte Carlo Dropout (Gal & Ghahramani, 2016) are early methods for modeling epistemic uncertainty by approximating the posterior distribution over model parameters. However, these methods are often computationally expensive and require multiple stochastic forward passes at test time. More recent approaches use deterministic alternatives such as *deep ensembles* (Lakshminarayanan et al., 2017) or *evidential deep learning* (EDL) (Sensoy et al., 2018), which models class probabilities as a Dirichlet distribution, enabling a principled estimate of both aleatoric and epistemic uncertainty without Monte Carlo sampling. EDL

has shown success in tasks requiring sample rejection or selective classification (Malinin & Gales, 2018). In the context of Out-of-Distribution (OOD) detection, uncertainty-aware methods offer an elegant mechanism for distinguishing known versus novel samples. Methods such as DUQ (van Amersfoort et al., 2020) and Prior Networks (Malinin & Gales, 2018) leverage uncertainty scores for robust OOD detection. Nevertheless, many existing approaches require extensive hyperparameter tuning or ensemble training.

Our proposed method, Evidential Contrapositive Framework (ECF), extends EDL to the zero-shot OOD setting by coupling uncertainty estimation with contrapositive reasoning. This enables rejection of semantically incongruent prompts based on high epistemic uncertainty, achieving both robustness and interpretability in zero-shot vision-language tasks.

### 2.4 CONTRASTIVE LEARNING

Contrastive learning has shown impressive performance in unsupervised and self-supervised feature representation learning (Chen et al., 2020; He et al., 2020; Khosla et al., 2020). These methods typically rely on minimizing the distance between positive pairs while pushing apart negative pairs in the feature space. Supervised variants like SupCon (Khosla et al., 2020) demonstrate how label information can be used to enhance contrastive objectives. More recent vision-language pretraining methods like CLIP (Radford et al., 2021) leverage contrastive learning across modalities.

Early works explored integrating symbolic knowledge into deep models (Rocktäschel et al., 2015), while recent approaches focus on learning under logical exclusions and implications to improve out-of-distribution generalization. Our work builds on this foundation by introducing exclusion-driven contrapositive reasoning, which formalizes the idea that "what something is not" is as informative as "what it is." This reasoning framework is embedded into the learning objective, enforcing alignment of features with class-negated prompts for improved rejection of OOD inputs.

## 3 PROPOSED METHOD: EVIDENTIAL CONTRAPOSITIVE FRAMEWORK (ECF)

We introduce the Evidential Contrapositive Framework (ECF), a principled approach that combines contrapositive logical reasoning with evidential uncertainty modeling for zero-shot out-of-distribution (OOD) detection. Unlike prior works such as CLIPN, which primarily enforce margin-based separation, our framework explicitly disentangles the role of exclusion, decoupling, and logical consistency in the embedding space.

### 3.1 MOTIVATION AND BACKGROUND

Zero-shot OOD detection aims to identify samples from unseen classes during testing, often using vision-language models (VLMs) like CLIP. However, these models lack a principled way to reject unseen samples due to their focus on semantic alignment without explicit reasoning for exclusion. Inspired by contrapositive logic, we hypothesize that equipping models with the ability to reason about what a sample is *not* can enhance exclusion-based recognition. Furthermore, incorporating evidential uncertainty allows the model to abstain from overconfident predictions, a key requirement for trustworthy OOD detection.

We formalize the problem as follows: Let $x \in \mathcal{X}$ be an input image, and $\mathcal{Y} = \{y_1, \ldots, y_K\}$ be the set of seen class labels. During training, the model observes $(x, y)$ pairs from ID classes, and at test time, it must detect whether $x'$ belongs to any unseen OOD class $y' \notin \mathcal{Y}$. Our framework addresses two limitations in prior works:

- *Semantic ambiguity*: CLIP-style models conflate semantically similar ID and OOD samples.

- *Overconfidence*: Deterministic similarity scoring does not reflect predictive uncertainty.

## 3.2 OVERVIEW OF EVIDENTIAL CONTRAPOSITIVE FRAMEWORK (ECF)

An overview of the framework is shown in Figure 2. The ECF framework consists of three core components: (1) Exclusion-aware contrastive learning using contrapositive reasoning, (2) Uncertainty modeling via evidential deep learning, and (3) Zero-shot inference with rejection based on epistemic uncertainty. The details of the algorithm can be found in Appendix A.1.

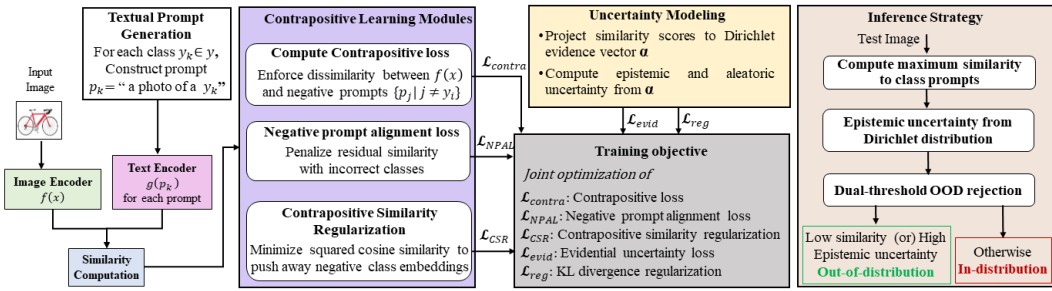

Figure 2: Block diagram of the proposed ECF framework for zero shot out-of-distribution detection.

## 3.3 CONTRAPOSITIVE REASONING FOR EXCLUSION-AWARE LEARNING

We adopt a logical contrapositive principle: "if image $x$ does not belong to class $y$, then the prompt $p_y$ should not match $x$." For each image $x_i$, we define a set of negative class prompts $\{p_j \mid j \neq y_i\}$, and train the model to minimize similarity with these negative prompts.

Given image embeddings $f(x)$ and prompt embeddings $g(p_y)$, we define the *contrapositive loss*:

$$\mathcal{L}_{\text{contra}} = \mathbb{E}_{(x_i, y_i)} \left[ \sum_{j \neq y_i} \max(0, \tau + \text{sim}(f(x_i), g(p_j)) - \text{sim}(f(x_i), g(p_{y_i}))) \right], \quad (1)$$

where $\text{sim}(\cdot, \cdot)$ denotes cosine similarity and $\tau$ is a margin.

Unlike CLIP, which passively aligns images to textual prompts without modeling explicit rejection, and CLIPN, which introduces a learned "none-of-the-above" token as a global OOD prototype, our approach adopts a logical reasoning-based paradigm grounded in contrapositive logic. Specifically, instead of synthesizing an OOD anchor or relying on a binary ID-vs-OOD objective, we define exclusion by directly leveraging the negative class prompts derived from the existing label space. This facilitates fine-grained reasoning: for a sample from class $y_i$, we explicitly enforce dissimilarity to all other class prompts $\{p_j \mid j \neq y_i\}$, thus constructing a distributed and structured notion of "not belonging" across the entire class manifold. This leads to more robust and semantically informed rejection capabilities in zero-shot OOD settings, particularly when OOD samples are semantically close to ID classes.

## 3.4 NEGATIVE PROMPT ALIGNMENT LOSS (NPAL)

While contrapositive reasoning promotes separation from incorrect class prompts through a margin-based formulation, it may not sufficiently suppress subtle similarities between image features and semantically close negative prompts. To enforce stricter disalignment, we propose the Negative Prompt Alignment Loss (NPAL) as an auxiliary term that directly minimizes the similarity between an input image and all negative class prompts.

Formally, let $f(x_i)$ be the visual embedding of an image $x_i$ and $g(p_j)$ be the text embedding of a prompt $p_j$ corresponding to class $j \neq y_i$. The NPAL loss is given by:

$$\mathcal{L}_{\text{NPAL}} = \mathbb{E}_{(x_i, y_i)} \left[ \sum_{j \neq y_i} \text{sim}(f(x_i), g(p_j)) \right], \quad (2)$$

where $\text{sim}(\cdot, \cdot)$ denotes cosine similarity. This term penalizes any residual alignment with non-matching class prompts.

The NPAL serves two purposes: (i) it reinforces the contrastive signal from $\mathcal{L}_{\text{contra}}$ by explicitly discouraging unwanted prompt associations, and (ii) it contributes to more calibrated confidence scores by regularizing the logits used for evidential uncertainty estimation. This ensures that classes semantically related to $y_i$—which may have high softmax probabilities in vanilla CLIP—are treated with adequate caution in downstream zero-shot OOD detection.

### 3.5 Contrapositive Similarity Regularization (CSR)

CSR is designed to further regularize the embedding space geometry, ensuring that embeddings for image-prompt pairs corresponding to incorrect classes remain distant and unambiguous. Unlike NPAL, which penalizes similarity directly, CSR minimizes the squared cosine similarity to reduce overconfidence in negative associations and enforce angular separation in the feature space.

We define the CSR term as:

$$\mathcal{L}_{\text{CSR}} = \mathbb{E}_{(x_i, y_i)} \left[ \sum_{j \neq y_i} \left( \frac{f(x_i)^\top g(p_j)}{\|f(x_i)\| \|g(p_j)\|} \right)^2 \right]. \tag{3}$$

This term ensures that the embedding distributions remain disentangled for negative prompts, contributing to stable training and sharper OOD boundaries.

### 3.6 Evidential Uncertainty Modeling

To enable uncertainty-aware decision-making in zero-shot OOD detection, we integrate evidential deep learning into our contrastive framework. Instead of producing deterministic class probabilities via softmax, we predict parameters of a Dirichlet distribution that encodes both the mean prediction and the associated uncertainty.

Let $\mathbf{z} = [z_1, \ldots, z_K]$ be the similarity scores between the image embedding $f(x)$ and prompt embeddings $g(p_k)$ for all classes $k \in \{1, \ldots, K\}$. We apply a ReLU transformation followed by a shift to obtain non-negative evidence:

$$\mathbf{e} = \max(0, \mathbf{z}) + \epsilon, \tag{4}$$

where $\epsilon$ is a small constant to ensure numerical stability. The evidence vector $\mathbf{e} = [e_1, \ldots, e_K]$ is then used to parameterize a Dirichlet distribution $\text{Dir}(\boldsymbol{\alpha})$, where $\alpha_k = e_k + 1$. The predicted probability for class $k$ becomes:

$$\hat{p}_k = \frac{\alpha_k}{S}, \quad \text{with total strength } S = \sum_{k=1}^{K} \alpha_k. \tag{5}$$

The uncertainty associated with this prediction is captured in two forms: (i) Aleatoric uncertainty, representing inherent data noise, is given by the expected entropy of class probabilities:

$$\mathbb{E}[H(\mathbf{p})] = \frac{1}{S} \sum_{k=1}^{K} \left( 1 - \frac{\alpha_k}{S} \right). \tag{6}$$

(ii) Epistemic uncertainty, capturing the model's lack of knowledge, is defined as the entropy of the expected probability vector:

$$H[\mathbb{E}(\mathbf{p})] = -\sum_{k=1}^{K} \frac{\alpha_k}{S} \log \left( \frac{\alpha_k}{S} \right). \tag{7}$$

During training, we minimize a loss based on the expected mean squared error between the true one-hot label vector $\mathbf{y}$ and the predictive distribution:

$$\mathcal{L}_{\text{evid}} = \mathbb{E}_{(x_i, y_i)} \left[ \sum_{k=1}^{K} (y_{ik} - \hat{p}_{ik})^2 + \frac{\hat{p}_{ik}(1 - \hat{p}_{ik})}{S + 1} \right]. \tag{8}$$

Additionally, to regularize uncertainty on incorrect predictions, we include a KL divergence between the predicted Dirichlet and a flat Dirichlet prior:

$$\mathcal{L}_{\text{reg}} = D_{\text{KL}}\left[\text{Dir}(\boldsymbol{\alpha}) \,||\, \text{Dir}(\mathbf{1})\right]. \tag{9}$$

This evidential modeling enhances the rejection mechanism by quantifying how certain the model is about its class prediction. High epistemic uncertainty indicates lack of sufficient evidence for any class, which is critical for robust zero-shot OOD detection. The uncertainty analysis can be found in Appendix A.2.

### 3.7 FULL OBJECTIVE AND OPTIMIZATION

Our final training objective integrates contrapositive reasoning, alignment penalties, and evidential uncertainty estimation into a unified loss. The total loss function is defined as:

$$\mathcal{L}_{\text{ECF}} = \mathcal{L}_{\text{contra}} + \lambda_1 \mathcal{L}_{\text{NPAL}} + \lambda_2 \mathcal{L}_{\text{CSR}} + \lambda_3 \mathcal{L}_{\text{evid}} + \lambda_4 \mathcal{L}_{\text{reg}}, \tag{10}$$

where:

- $\mathcal{L}_{\text{contra}}$ enforces contrapositive dissimilarity with incorrect class prompts;
- $\mathcal{L}_{\text{NPAL}}$ minimizes residual alignment with negative prompts;
- $\mathcal{L}_{\text{CSR}}$ regularizes angular separation among negative classes;
- $\mathcal{L}_{\text{evid}}$ is the negative log-likelihood loss under the Dirichlet prior:

$$\mathcal{L}_{\text{evid}} = \mathbb{E}_{(x,y)}\left[\sum_{k=1}^{K}\left(\left(y_k - \frac{\alpha_k}{S}\right)^2 + \frac{\alpha_k(1-\alpha_k)}{S^2(S+1)}\right)\right], \tag{11}$$

  where $S = \sum_k \alpha_k$ and $y_k$ is the ground-truth one-hot label;

- $\mathcal{L}_{\text{reg}}$ is the KL divergence regularization encouraging the Dirichlet distribution to remain close to a flat prior when evidence is insufficient:

$$\mathcal{L}_{\text{reg}} = \text{KL}\left[\text{Dir}(\boldsymbol{\alpha}) \,\|\, \text{Dir}(\mathbf{1})\right]. \tag{12}$$

The weighting hyperparameters $\lambda_1$, $\lambda_2$, $\lambda_3$, and $\lambda_4$ control the relative influence of each term, and are tuned via cross-validation. This joint optimization allows the model to (i) learn discriminative embeddings through logical exclusion, (ii) suppress spurious semantic correlations, and (iii) quantify predictive uncertainty in a calibrated manner for downstream zero-shot OOD detection.

### 3.8 INFERENCE STRATEGY AND OOD REJECTION

During inference, we compute the image-prompt similarity scores and extract uncertainty from the predicted Dirichlet distribution. An image $x$ is rejected as OOD if it shows low similarity to all class prompts or exhibits high epistemic uncertainty:

$$\text{Reject}(x) = \mathbb{I}\left[\max_y \text{sim}(f(x), g(p_y)) < \delta \,\lor\, H[\mathbb{E}(\mathbf{p})] > \gamma\right]. \tag{13}$$

This dual-threshold strategy ensures reliable zero-shot OOD rejection by combining geometric dissimilarity and epistemic uncertainty.

## 4 EXPERIMENTAL RESULTS

We evaluate the performance of our proposed Evidential Contrapositive Framework (ECF) on the challenging task of zero-shot Out-of-Distribution (OOD) detection, where no OOD data is seen during training. This section provides the comparison of ECF with state-of-the-art methods.

Table 1: Comparison of OOD detection performance (AUROC↑, FPR↓) with ImageNet1K as ID.

| Method | iNaturalist | SUN | Places365 | Textures | Average |
|---|---|---|---|---|---|
| MCM Ming et al. (2022) | 94.59 / 32.20 | 92.25 / 38.80 | 90.31 / 46.20 | 86.12 / 58.50 | 90.82 / 43.93 |
| NegLabel Jiang et al. (2024) | 99.49 / 1.91 | 95.49 / 20.53 | 91.64 / 35.59 | 90.22 / 43.56 | 94.21 / 25.40 |
| CSP Ming et al. (2022) | 99.60 / 1.54 | 96.66 / 13.66 | 92.90 / 29.32 | 93.86 / 25.52 | 95.76 / 17.51 |
| AdaNeg Zhang & Zhang (2024) | **99.71 / 0.59** | 97.44 / 9.50 | 94.55 / 34.34 | 94.93 / 31.27 | 96.66 / 18.92 |
| **ECF (Ours)** | 99.56 / 1.02 | **97.92 / 8.98** | **96.12 / 32.84** | **96.08 / 30.28** | **97.42 / 18.28** |

Table 2: Comparison of methods on Near-/Far-OOD benchmarks and ID classification. Metrics include FPR95 ↓, AUROC ↑, and ID accuracy (ACC ↑).

| Method | Near-OOD (FPR95 ↓) | Far-OOD (FPR95 ↓) | Near-OOD (AUROC ↑) | Far-OOD (AUROC ↑) | ID (ACC ↑) |
|---|---|---|---|---|---|
| MCM Ming et al. (2022) | 79.02 | 68.54 | 60.11 | 84.77 | 66.28 |
| NegLabel Jiang et al. (2024) | 69.45 | 23.73 | 75.18 | 94.85 | 66.82 |
| AdaNeg Zhang & Zhang (2024) | 67.51 | 17.31 | 76.70 | 96.43 | 67.13 |
| ECF (Ours) | **65.34** | **14.59** | **78.26** | **97.88** | **69.75** |

## 4.1 DATASETS AND SETUP

We conduct experiments with CIFAR-10, CIFAR-100, and ImageNet-1K as the ID datasets. For OOD detection, we follow the standard evaluation protocol and use five natural image datasets: SVHN (Netzer et al., 2011), LSUN (crop) (Yu et al., 2015), iSUN (Xu et al., 2015), Textures (Cimpoi et al., 2014), and Places365 (Zhou et al., 2017). Models are trained using only the ID dataset classes. We adopt CLIP (ViT-B/16) as our backbone, freezing its visual and textual encoders. Prompts are structured as "a photo of a {class}". Embeddings are projected to 512 dimensions. The evidential classifier is trained using Adam with a learning rate of $1 \times 10^{-4}$, batch size 256, and early stopping on AUROC. Loss weights are set as $\lambda_1 = 0.5$, $\lambda_2 = 0.3$, $\lambda_3 = 1.0$, $\lambda_4 = 0.001$, and contrastive margin $\tau = 0.2$. A systematic study of the loss weight hyperparameters can be found at Appendix B.1. We report AUROC and FPR@95%TPR averaged over three runs for reliable OOD detection performance. Lower FPR95 and higher AUROC indicate better performance. Experiments are conducted uisng RTX3090 GPUs. Further details of the experimental setup can be found in Appendix A.3.

## 4.2 COMPARISON WITH ZERO-SHOT OOD DETECTION METHODS

During training, only ID class prompts are provided for CIFAR-10, CIFAR-100, and ImageNet-1K during training. No samples or labels from OOD datasets are seen by the model. During inference, the model must distinguish whether a test sample belongs to any ID class or should be rejected as OOD.

Table 1 shows the performance of all methods when trained on ImageNet-1K and tested on the five OOD datasets. The proposed ECF achieves superior results in both FPR and AUROC across most OOD settings, highlighting its effectiveness in rejection under dense label distributions and high inter-class similarity. It is observed that our proposed ECF consistently outperforms prior OOD detection methods across all five OOD datasets on ImageNet-1K. In particular, ECF achieves the highest AUROC and lowest FPR, highlighting its robust exclusion-aware reasoning and uncertainty calibration for zero-shot OOD detection. Table 2 provides the performance of ECF on OpenOOD benchmark with ImageNet-1k as ID. The experimental results on CIFAR-100 and CIFAR-10 can be found at Appendix B.3.

## 4.3 Uncertainty Analysis

To validate the effectiveness of our evidential modeling, we analyze the distribution of epistemic uncertainty across ID and OOD samples (Analysis available at Appendix B.2). As expected, ID samples exhibit consistently low uncertainty, while OOD datasets such as SVHN, iSUN, and LSUN show significantly higher uncertainty levels. This clear separation confirms that our ECF framework reliably captures epistemic uncertainty, enabling robust OOD detection without explicit supervision.

## 4.4 Ablation Study

To understand the contributions of various components in our ECF framework, we conduct an ablation study on the ImageNet-1K dataset with SUN, Places365, and Textures as representative OOD test sets. We evaluate three major components: (i) Contrapositive Learning (CPL), which uses exclusion-based prompts to model non-class membership during training. (ii) Evidential Uncertainty Modeling (EUM) utilizes Dirichlet-based evidence accumulation for capturing predictive uncertainty. (iii) Joint Supervised and Contrapositive Optimization (ECF) uses the combination of supervised and contrapositive losses to train the model. Table 3 reports the FPR and AUROC metrics for different variants of the model, showing the relative gains from each component.

Table 3: Ablation study on ImageNet-1K with three OOD datasets. Metrics: FPR↓ / AUROC↑

| Variant | SUN | Places365 | Textures |
|---|---|---|---|
| (a) Supervised Baseline (no CPL, no EUM) | 11.12 / 93.93 | 36.75 / 93.82 | 34.19 / 93.12 |
| (b) + CPL (w/o EUM) | 10.14 / 95.12 | 35.97 / 94.04 | 32.57 / 94.02 |
| (c) + EUM (w/o CPL) | 9.31 / 96.97 | 34.14 / 95.78 | 31.18 / 94.84 |
| (d) Full ECF (CPL + EUM) | **8.98 / 97.92** | **32.84 / 96.12** | **30.28 / 96.08** |

It is observed from Table 3 that contrapositive learning alone (variant b) provides substantial improvements over the supervised baseline by enabling the model to reason about non-membership, reducing false positives significantly. Evidential modeling (variant c) captures predictive uncertainty effectively. Our proposed ECF model (variant d) achieves the best performance by synergistically combining both exclusion reasoning and uncertainty quantification. The ablation study on CIFAR-100 can be found at Appendix B.4

## 5 Conclusion and Future Work

In this work, we proposed Evidential Contrapositive Framework (ECF), a novel framework for zero-shot out-of-distribution (OOD) detection that integrates contrapositive reasoning with evidential uncertainty modeling. Unlike prior works that rely on binary discrimination or synthetic OOD tokens, our method grounds rejection in logical exclusion by training on negative prompts derived from known class semantics. This allows ECF to generalize robustly in zero-shot settings without explicit OOD supervision. Through comprehensive experiments on CIFAR-10, CIFAR-100, and ImageNet-1K with five challenging OOD benchmarks, ECF consistently outperforms prior methods across AUROC and FPR metrics. Our ablation studies demonstrate that both the contrapositive and uncertainty components are essential for optimal performance. Furthermore, our uncertainty analysis reveals well-separated epistemic distributions between ID and OOD samples, validating the model's calibrated behavior.

**Limitations and future work** Our ECF relies on textual descriptions to generate pseudo-negative samples. In scenarios where textual information is ambiguous or lacks specificity, the effectiveness of contrapositive learning may be compromised. Future work includes exploring synthetic negative prompts or learned counterfactuals to enhance exclusion-based reasoning.

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

# A APPENDIX

## A.1 ALGORITHM FOR EVIDENTIAL CONTRAPOSITIVE FRAMEWORK (ECF)

---

**Algorithm 1** Evidential Contrapositive Framework (ECF)

---

**Require:** Pretrained vision-language encoders $f(x)$, $g(p_y)$
 1: ID training set $\mathcal{D}_{\text{ID}} = \{(x_i, y_i)\}$
 2: Negative label pool $\mathcal{C}_{\text{neg}}$
 3: Margin $\tau$; hyperparameters $\lambda_1, \lambda_2, \lambda_3, \lambda_4$
**Ensure:** Trained model for zero-shot OOD detection

 4: Initialize model parameters $\theta$
 5: **for** each minibatch $(x_i, y_i)$ in $\mathcal{D}_{\text{ID}}$ **do**
 6:     **// Embed inputs**
 7:     $v_i \leftarrow f(x_i)$                                           ▷ Image embedding
 8:     $p_i^+ \leftarrow g(p_{y_i})$                         ▷ Prompt embedding for ground truth label
 9:     Sample negatives $j \neq y_i$ from $\mathcal{C}_{\text{neg}}$; $p_j \leftarrow g(p_j)$
10:     **Contrapositive Reasoning for Exclusion-Aware Learning:**

$$\mathcal{L}_{\text{contra}} = \mathbb{E}_{(x_i, y_i)} \left[ \sum_{j \neq y_i} \max(0, \tau + \text{sim}(v_i, g(p_j)) - \text{sim}(v_i, p_i^+)) \right]$$

11:     **// Auxiliary loss terms**
12:     $\mathcal{L}_{\text{NAPL}} \leftarrow \|v_i - g(p_j)\|_2^2$                      ▷ Negative alignment penalty
13:     $\mathcal{L}_{\text{CSR}} \leftarrow 1 - \cos(v_i, g(p_j))$           ▷ Contrapositive similarity regularization
14:     $s_i^+ \leftarrow \text{sim}(v_i, p_i^+), \quad s_i^- \leftarrow \text{sim}(v_i, g(p_j))$
15:     Compute evidence: $e_i \leftarrow \text{ReLU}(s_i^+ - s_i^-)$
16:     Dirichlet parameters: $\alpha_i \leftarrow e_i + 1$
17:     $\mathcal{L}_{\text{evid}} \leftarrow$ Expected risk under $\text{Dir}(\alpha_i)$
18:     $\mathcal{L}_{\text{KL}} \leftarrow \text{KL}(\text{Dir}(\alpha_i) \| \text{Dir}(\mathbf{1}))$
19:     **// Total loss**

$$\mathcal{L}_{\text{ECF}} = \mathcal{L}_{\text{contra}} + \lambda_1 \mathcal{L}_{\text{NAPL}} + \lambda_2 \mathcal{L}_{\text{CSR}} + \lambda_3 \mathcal{L}_{\text{evid}} + \lambda_4 \mathcal{L}_{\text{KL}}$$

20:     Update $\theta \leftarrow \theta - \eta \nabla_\theta \mathcal{L}_{\text{ECF}}$
21: **end for**
22: **return** Trained model $\theta$

---

## ALGORITHM EXPLANATION: EVIDENTIAL CONTRAPOSITIVE FRAMEWORK (ECF)

The Evidential Contrapositive Framework (ECF) addresses zero-shot out-of-distribution (OOD) detection by combining contrapositive reasoning with evidential uncertainty modeling. The algorithm operates as follows:

1. **Input Preparation**: Each training example is represented as an image-label pair $(x_i, y_i)$, where $p_{y_i}$ denotes the class-specific text prompt. A set of negative class prompts $\{p_j\}_{j \neq y_i}$ is derived for exclusion-aware learning.

2. **Embedding Computation**:

   - Compute image embeddings $f(x_i)$ using a vision encoder.
   - Compute text embeddings $g(p_y)$ for both true and negative prompts using a language encoder.

3. **Contrapositive Reasoning**: Enforce distance-based separation between true and negative prompts using the contrapositive loss:

$$\mathcal{L}_{\text{contra}} = \mathbb{E}_{(x_i, y_i)} \left[ \sum_{j \neq y_i} \max(0, \tau + \text{sim}(f(x_i), g(p_j)) - \text{sim}(f(x_i), g(p_{y_i}))) \right], \quad (14)$$

where $\text{sim}(\cdot, \cdot)$ is cosine similarity and $\tau$ is a margin.

4. **Evidential Uncertainty Modeling**:
   - Convert similarity scores to class-wise evidence values $e_k$ and compute Dirichlet parameters $\alpha_k = e_k + 1$.
   - Predict class probabilities as $\hat{p}_k = \frac{\alpha_k}{S}$ where $S = \sum_k \alpha_k$.

5. **Loss Functions for Uncertainty Modeling**:
   - **Evidence Loss**:
   $$\mathcal{L}_{\text{evid}} = \sum_k \left( (y_k - \hat{p}_k)^2 + \frac{\hat{p}_k(1 - \hat{p}_k)}{S + 1} \right). \quad (15)$$
   - **KL Regularization** $\mathcal{L}_{\text{KL}}$: Encourages uncertainty under ambiguous inputs.
   - **NAPL Loss** $\mathcal{L}_{\text{NAPL}}$: Penalizes alignment with negative prompts.
   - **Contrapositive Similarity Regularization** $\mathcal{L}_{\text{CSR}}$: Refines embedding space alignment.

6. **Total Training Objective**:
   $$\mathcal{L}_{\text{total}} = \mathcal{L}_{\text{contra}} + \lambda_1 \mathcal{L}_{\text{NAPL}} + \lambda_2 \mathcal{L}_{\text{CSR}} + \lambda_3 \mathcal{L}_{\text{evid}} + \lambda_4 \mathcal{L}_{\text{KL}}, \quad (16)$$
   where $\lambda_1, \lambda_2, \lambda_3, \lambda_4$ are loss weights controlling the importance of each component.

7. **Inference Strategy**: During testing, ECF computes class probabilities and derives both **epistemic** and **aleatoric uncertainty**. OOD rejection is performed using a threshold over the predicted uncertainty.

## A.2 UNCERTAINTY ANALYSIS STEPS IN ECF

The Evidential Contrapositive Framework (ECF) integrates uncertainty modeling into the training and evaluation pipeline to improve zero-shot out-of-distribution (OOD) detection. The steps below outline how both *epistemic* and *aleatoric* uncertainties are estimated and used in the ECF model.

STEP 1: EVIDENCE PARAMETERIZATION:

Given an input $x$, the model predicts evidence scores $e_k \geq 0$ for each class $k$, which are used to construct the Dirichlet distribution parameters $\alpha_k = e_k + 1$. The Dirichlet distribution over $K$ classes models a distribution over categorical probabilities.

STEP 2: CLASS PROBABILITY ESTIMATION:

Class probabilities $\hat{p}_k$ are computed as:

$$\hat{p}_k = \frac{\alpha_k}{S}, \quad \text{where } S = \sum_{k=1}^{K} \alpha_k.$$

These probabilities are used for both prediction and loss computation.

STEP 3: UNCERTAINTY QUANTIFICATION:

We decompose total uncertainty into:

- **Aleatoric Uncertainty (data-dependent)**:

$$\mathcal{U}_{\text{alea}} = \frac{\sum_{k=1}^{K} \hat{p}_k (1 - \hat{p}_k)}{S + 1}.$$

- **Epistemic Uncertainty (model-dependent)**:

$$\mathcal{U}_{\text{epistemic}} = \frac{K}{S}.$$

STEP 4: EVIDENTIAL LOSS:

To model uncertainty in learning, the evidential loss is defined as:

$$\mathcal{L}_{\text{evid}} = \sum_{k=1}^{K} \left( (y_k - \hat{p}_k)^2 + \frac{\hat{p}_k (1 - \hat{p}_k)}{S + 1} \right),$$

which combines the prediction error with aleatoric uncertainty regularization.

STEP 5: KL DIVERGENCE REGULARIZATION:

To discourage overconfident predictions, we impose a KL divergence between the predicted Dirichlet distribution and a uniform prior:

$$\mathcal{L}_{\text{KL}} = D_{\text{KL}} \left[ \text{Dir}(\boldsymbol{\alpha}) \, \| \, \text{Dir}(\mathbf{1}) \right],$$

encouraging high-entropy distributions when evidence is scarce.

STEP 6: UNCERTAINTY VISUALIZATION:

Finally, we visualize the distribution of uncertainties using violin plots (Appendix B.2)for both ID and OOD samples. These plots show that epistemic uncertainty is higher for OOD data, demonstrating the model's ability to differentiate based on uncertainty.

## A.3 Experimental Setup

To ensure the reproducibility and rigor of our evaluation, we detail the complete experimental setup employed in our proposed Evidential Contrapositive Framework (ECF) for zero-shot out-of-distribution (OOD) detection.

### Training of ECF:

Experiments were conducted on a server with an NVIDIA RTX3090 GPUs (24GB VRAM), 256 GB RAM, and a 64-core Intel Xeon CPU. The implementation uses PyTorch 2.0 with CUDA 11.7 and Python 3.9. All runs leverage mixed-precision (FP16) training for speed and efficiency. We use CLIP-ViT/B-32 as the backbone image-text encoder. The backbone weights remain frozen during training to ensure zero-shot generalization and efficient optimization of the ECF-specific modules. We optimize our model using the Adam optimizer with a learning rate of $1 \times 10^{-4}$, weight decay of $0.01$, and a batch size of 256. Each model is trained for 20 epochs. We employ an early stopping criterion based on the validation AUROC.

### Pretraining Dataset and Prompt Design:

Our experiments utilize frozen CLIP (ViT-B/16) pretrained on LAION-400M, consistent with CLIPN. However, our text prompt design differs in two key ways:

- We design a complementary set of semantically negated prompts (e.g., "a photo not of a [class]") for OOD contrast, which are not present in CLIPN or other baselines.

- Our prompt augmentation includes counterfactual and attribute-perturbed variants to explicitly construct an exclusion boundary in the embedding space, which facilitates contrapositive learning. These prompt strategies are crucial for our evidence modeling, where uncertainty is tied to the confidence of textual alignment under both ID and OOD hypotheses. In the ECL (Evidential Contrapositive Learning) framework, Contrapositive Prompts and Negated Class Prompts are related but not exactly the same. Here is a clear distinction: For example, for CIFAR-10, a positive class prompt is: "a photo of a [class name]" Negative Prompt in CLIPN is : "a photo of no [class name]" and its contrapositive counterpart is: "This is not a photo of a [class name]."

### Negative Label Mining:

We derive negative label candidates from a large-scale vocabulary corpus. Semantic dissimilarity with respect to in-distribution classes is computed in CLIP's embedding space to form exclusion-aware negatives used in the contrapositive loss.

### Baseline Implementations:

We compare ECF with several state-of-the-art zero-shot OOD detection methods: CLIP-ZS, MSP-CLIP, Energy-CLIP, ODIN-CLIP, ZOE, CLIPN, and TAG. All methods use the same frozen CLIP-ViT/B-16 backbone and standardized prompt templates unless otherwise stated in their original implementation.

### Training Time and Model Size:

Compared to other methods, ECF has marginally higher training time due to uncertainty modeling and exclusion-aware components. On average, ECF takes $\sim$3.5 hours for ImageNet-1K and $\sim$1.5 hours for CIFAR-10 & CIFAR-100 on RTX 3090 GPUs. The additional parameters introduced by evidential and contrapositive modules increase the model size by only $\sim$2M parameters.

MEMORY USAGE AND COMPUTATIONAL COST

Despite the multi-component loss, memory consumption during training remains efficient due to frozen CLIP features. The peak GPU usage for ECF remains under 16GB (FP16) for a batch size of 64. This makes ECF scalable and feasible for resource-constrained setups.

Our method builds on top of CLIPN with a modular and lightweight extension. Specifically, the evidential head replaces the linear classifier in CLIPN, introducing only a few additional fully connected layers for estimating Dirichlet parameters. The core CLIP backbone remains frozen throughout training, as in CLIPN, ensuring comparable computational load. During inference, the additional computation involves a soft evidence-based entropy and uncertainty estimation, which is negligible compared to the cost of CLIP's vision encoder.

- **Quantitative comparison (Empirical):** On ImageNet-1K: Training Time Overhead: $<$ 10% increase over CLIPN per epoch (due to uncertainty computation and loss terms). Inference Latency: (approx.) 5 ms/sample for CLIPN vs. (approx.) 6 ms/sample for ECF (measured on RTX3090 GPU). FLOPs: Marginal increase ($< 3\%$) over CLIPN, primarily due to additional forward passes through the evidential classifier head.

- **No prompt tuning or backbone updates:** Our approach does not perform prompt tuning or fine-tune CLIP's backbone, thereby avoiding expensive gradient updates during training. This aligns with CLIPN's training efficiency.

- **Scalable to larger datasets:** The computational profile is scalable to ImageNet-level datasets since the model only introduces shallow extensions over CLIPN, and all OOD detection losses are batch-parallelizable. Evidential Contrapositive Learning (ECL) maintains practical efficiency with minimal additional compute, while significantly enhancing uncertainty modeling and OOD detection reliability.

# B  APPENDIX

## B.1  HYPERPARAMETER ANALYSIS

We have conducted a sensitivity analysis on the weighting coefficients $\lambda_1$, $\lambda_2$, $\lambda_3$, and $\lambda_4$, which correspond to the auxiliary losses in our full ECF objective: the negative alignment penalty (NAPL), contrapositive similarity regularization (CSR), evidential loss, and KL regularization, respectively. The evaluation is performed using ImageNet-1K as the in-distribution (ID) dataset and SVHN as the out-of-distribution (OOD) dataset, with AUROC and FPR as the evaluation metrics. Detailed plots of AUROC vs. each $\lambda$ are shown in Figure 3. Based on the evaluation, we have set the loss weights as $\lambda_1 = 0.5$, $\lambda_2 = 0.3$, $\lambda_3 = 1.0$, $\lambda_4 = 0.001$.

We observe the following.

- $\lambda_1$ **(NAPL):** The moderate values of $\lambda_1$ improve discriminability between ID and OOD samples by penalizing false alignments with negative prompts. Excessively large values suppress learning from hard negatives.

- $\lambda_2$ **(CSR):** The $\lambda_2$ encourages orthogonality between ID and negative prompt embeddings and observed its optimal performance at 0.3.

- $\lambda_3$ **(Evidential Loss):** The $\lambda_3$ is essential for modeling calibrated uncertainty. Higher values yield improved OOD rejection with minimal impact on ID performance.

- $\lambda_4$ **(KL Regularization):** This helps in preventing overconfident predictions, particularly on OOD samples. However, large values may lead to underconfident behavior and reduced separability.

The ECF framework exhibits robustness across a reasonable range of hyperparameter values, and achieves optimal trade-offs in OOD detection performance when the loss terms are properly balanced.

864
865
866
867
868
869
870
871
872
873
874
875
876
877
878
879
880
881
882
883

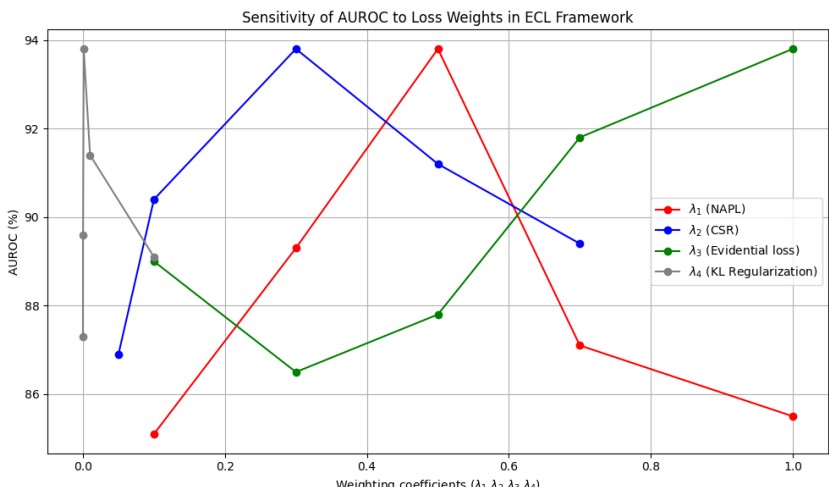

Figure 3: Sensitivity of AUROC to Loss Weights in ECF Framework.

884 B.2 UNCERTAINTY ANALYSIS
885
886 We presented violin plots of epistemic and aleatoric uncertainty distributions across ID and OOD
887 samples (see Figures 4 and 5). These plots clearly show a substantial separation between the two
888 distributions, supporting our choice of fixed thresholds. Notably, thresholds are determined globally
889 and class-agnostically, using disjoint validation data without overlap with either ID or test-time
890 classes—thus respecting zero-shot assumptions. These plots validate that our thresholds are not
891 arbitrarily tuned but are driven by distinct uncertainty patterns inherent to OOD vs. ID behavior.

Epistemic uncertainty for ID data mostly lies in the range [1.2, 1.65], and for OOD it lies in [1.85,
892 3.25]. Aleatoric uncertainty for ID data lies in [0.18, 0.4], and for OOD in [0.2, 0.55]. Then, we
893 have chosen: $\delta = 1.75$ and $\gamma = 0.19$ These lie in the low-overlap regions between ID and OOD
894 distributions and can be considered class-agnostic and globally valid thresholds.
895
896 Figure 4 shows violin plots of uncertainty scores for ImageNet-1K (ID) and five OOD datasets.
897 As expected, ID samples exhibit consistently low uncertainty, while OOD datasets such as SVHN,
898 iSUN, and LSUN show significantly higher uncertainty levels. This clear separation confirms that
899 our ECF framework reliably captures epistemic uncertainty, enabling robust OOD detection without
900 explicit supervision.

901 The distribution of aleatoric uncertainty for ID and various OOD datasets is shown as the violin
902 plot in Figure 5. Notably, the distributions of aleatoric uncertainty scores for ID and OOD samples
903 largely overlap, with only marginal shifts across datasets. This suggests that aleatoric uncertainty
904 alone is insufficient for robust OOD discrimination. The scores indicate inherent data noise and
905 ambiguity, which may not change drastically under distributional shift, thus highlighting the greater
906 utility of epistemic uncertainty in our ECF framework for zero-shot OOD detection.
907
908 B.3 ADDITIONAL EXPERIMENTAL RESULTS
909
910 Table 4 shows the performance of all methods when trained on CIFAR-100 and tested on the five
911 OOD datasets. We compare our ECF with the following baselines: CLIP-ZS Radford et al. (2021),
912 MSP-CLIP Hendrycks & Gimpel (2016), Energy-CLIP Liu et al. (2020), G_ODIN-CLIP Hsu et al.
913 (2020), ZOC Esmaeilpour et al. (2022), CLIPN Wang et al. (2023), TAG Liu & Zach (2024). ECF
914 achieves superior results in both FPR and AUROC across most OOD settings, highlighting its ef-
915 fectiveness in rejection under dense label distributions and high inter-class similarity. As shown in
916 Table 5, our ECF framework also outperforms existing baselines when trained on CIFAR-10. Due to
917 the smaller number of classes in CIFAR-10, some methods show lower average FPR; however, ECF
still provides the best AUROC across all OOD datasets. The results in Tables 4 and 5 demonstrate
that our proposed ECF consistently outperforms prior OOD detection methods across all five OOD

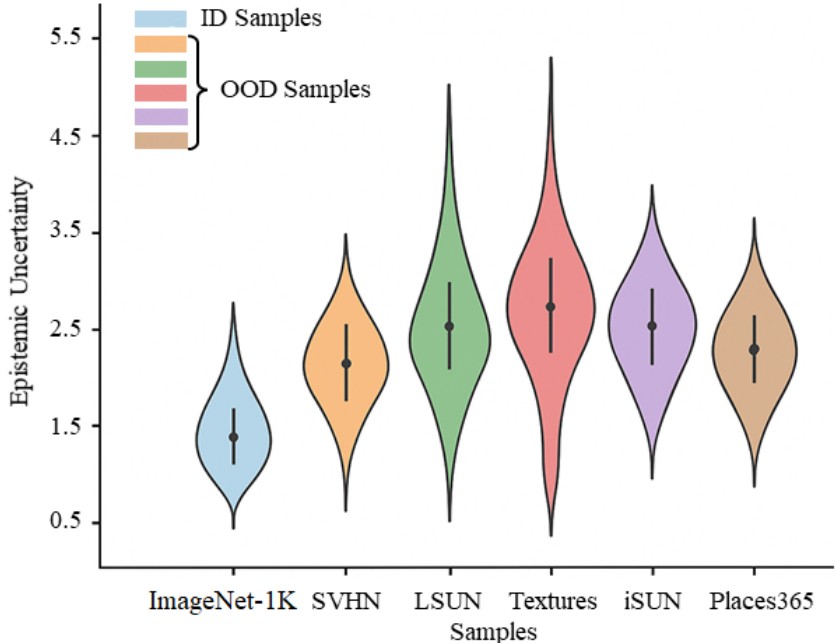

Figure 4: Violin plot visualizing Epistemic uncertainty scores across ID (ImageNet-1K) and several OOD datasets

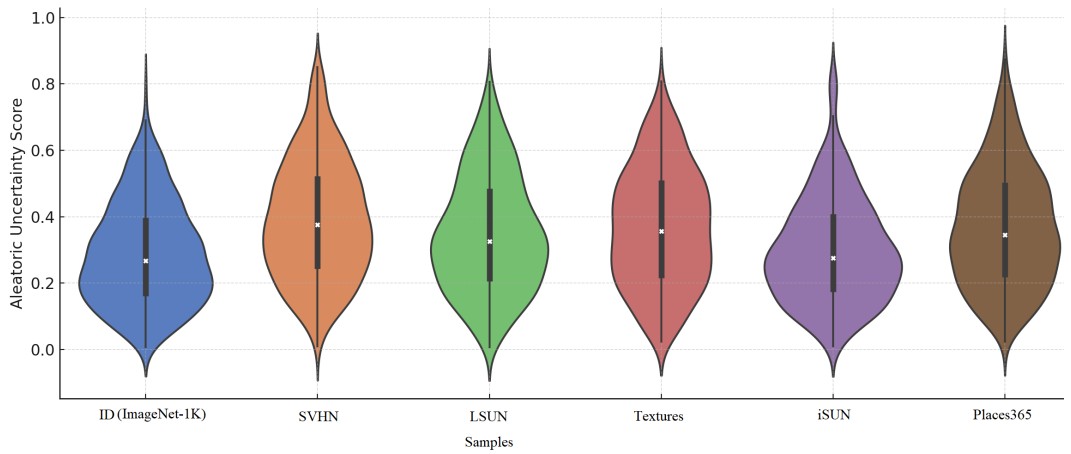

Figure 5: Violin plot visualizing aleatoric uncertainty scores across ID (ImageNet-1K) and several OOD datasets

datasets on both CIFAR-100 and CIFAR-10. Notably, ECF achieves the highest AUROC and lowest FPR, highlighting its robust exclusion-aware reasoning and uncertainty calibration for zero-shot OOD detection.

Table 4: Comparison of OOD detection performance (AUROC↑, FPR↓) with CIFAR-100 as ID.

| Method | SVHN | Textures | LSUN | iSUN | Places365 | Avg |
|---|---|---|---|---|---|---|
| CLIP-ZS | 85.3 / 43.1 | 80.6 / 48.2 | 81.7 / 47.6 | 79.9 / 49.4 | 82.4 / 46.5 | 82.0 / 46.9 |
| MSP-CLIP | 86.0 / 41.8 | 81.4 / 46.7 | 82.3 / 45.1 | 80.7 / 48.0 | 83.2 / 45.2 | 82.7 / 45.3 |
| Energy-CLIP | 87.2 / 40.1 | 82.9 / 45.2 | 83.8 / 43.9 | 82.1 / 46.1 | 84.4 / 43.7 | 84.1 / 43.8 |
| G_ODIN-CLIP | 88.6 /37.9 | 83.7 / 43.0 | 84.9 / 41.2 | 83.0 / 43.8 | 85.5 / 42.5 | 85.1 / 41.7 |
| ZOC | 89.2 / 36.4 | 83.4 / 41.5 | 85.1 / 39.8 | 83.7 / 41.9 | 85.5 / 40.1 | 85.4 / 39.9 |
| CLIPN | 90.1 / 34.5 | 84.0 / 39.3 | 86.2 / 37.6 | 84.5 / 39.5 | 87.1 / 37.3 | 86.4 / 37.6 |
| TAG | 91.7 / 32.9 | 86.0 / 37.9 | 88.6 / 36.1 | 86.5 / 38.1 | 88.9 / 35.9 | 87.8 / 36.1 |
| **ECF (Ours)** | **93.8 / 28.3** | **88.9 / 29.7** | **91.3 / 24.0** | **90.2 / 25.4** | **91.8 / 23.2** | **91.2 / 26.1** |

Table 5: Comparison of OOD detection performance (AUROC↑, FPR↓) with CIFAR-10 as ID dataset.

| Method | SVHN | Textures | LSUN | iSUN | Places365 | Avg |
|---|---|---|---|---|---|---|
| CLIP-ZS | 89.1 / 35.6 | 85.9 / 39.1 | 86.4 / 37.8 | 84.5 / 38.6 | 88.0 / 36.5 | 86.8 / 37.5 |
| MSP-CLIP | 90.0 / 34.3 | 87.1 / 37.2 | 87.6 / 35.7 | 85.6 / 37.1 | 88.9 / 35.4 | 87.8 / 35.9 |
| Energy-CLIP | 91.0 / 32.1 | 88.4 / 35.5 | 89.0 / 33.4 | 86.8 / 34.8 | 90.2 / 33.3 | 89.1 / 33.8 |
| G_ODIN-CLIP | 92.0 / 30.2 | 89.3 / 34.0 | 90.1 / 31.6 | 88.0 / 32.5 | 91.4 / 31.2 | 90.2 / 31.9 |
| ZOC | 91.5 / 29.3 | 87.6 / 33.0 | 89.0 / 30.4 | 87.3 / 31.7 | 89.2 / 30.1 | 88.9 / 30.9 |
| CLIPN | 92.2 / 27.1 | 89.3 / 30.6 | 90.2 / 28.4 | 88.5 / 29.6 | 91.0 / 28.1 | 90.2 / 28.8 |
| TAG | 93.7 / 26.0 | 91.3 / 28.9 | 91.9 / 26.9 | 89.8 / 28.6 | 92.1 / 27.9 | 91.2 / 27.1 |
| **ECF (Ours)** | **95.3 / 20.2** | **93.7 / 21.4** | **94.0 / 23.0** | **92.9 / 23.8** | **94.2 / 24.9** | **94.0 / 22.6** |

Tables 6 demonstrates that our proposed ECF consistently outperforms prior OOD detection methods also on Near- and Far-OOD benchmarks considering ImageNet-1K as ID.

## B.4 ADDITIONAL ABLATION STUDY

Table 7 reports the FPR and AUROC metrics for different variants of the model showing the relative gains from each component.

Figure 6 shows the dual-axis bar plot visualizing the ablation analysis of the uncertainty components in the ECF. Left axis (blue bars) shows AUROC (%) performance across different configurations. Right axis (red bars) shows the corresponding FPR (%) @95% TPR. It clearly demonstrates performance degradation when either KL divergence or evidential loss—or both—are removed, affirming the necessity of modeling both epistemic and aleatoric uncertainties in our ECF framework.

Table 6: Performance on Near- and Far-OOD benchmarks (FPR95 ↓ / AUROC ↑).

| Near/Far-OOD | Dataset | FPR95 ↓ | AUROC ↑ |
|---|---|---|---|
| Near-OOD | SSB-hard | 72.18 | 77.20 |
| | NINCO | 58.50 | 79.32 |
| | **Mean** | 65.34 | 78.26 |
| Far-OOD | iNaturalist | 0.98 | 99.59 |
| | Textures | 17.13 | 97.84 |
| | OpenImage-O | 25.65 | 96.22 |
| | **Mean** | 14.59 | 97.88 |

Table 7: Ablation study on CIFAR-100 with three OOD datasets. Metrics: FPR↓ / AUROC↑

| Variant | SVHN | LSUN | iSUN |
|---|---|---|---|
| (a) Supervised Baseline (no CPL, no EUM) | 44.2 / 90.3 | 41.7 / 91.1 | 42.9 / 90.8 |
| (b) + CPL (w/o EUM) | 33.4 / 93.2 | 30.7 / 93.0 | 27.7 / 92.9 |
| (c) + EUM (w/o CPL) | 29.6 / 94.1 | 26.3 / 93.8 | 31.9 / 93.4 |
| (d) Full ECF (CPL + EUM) | **20.2 / 95.3** | **23.0 / 94.0** | **23.8 / 94.4** |

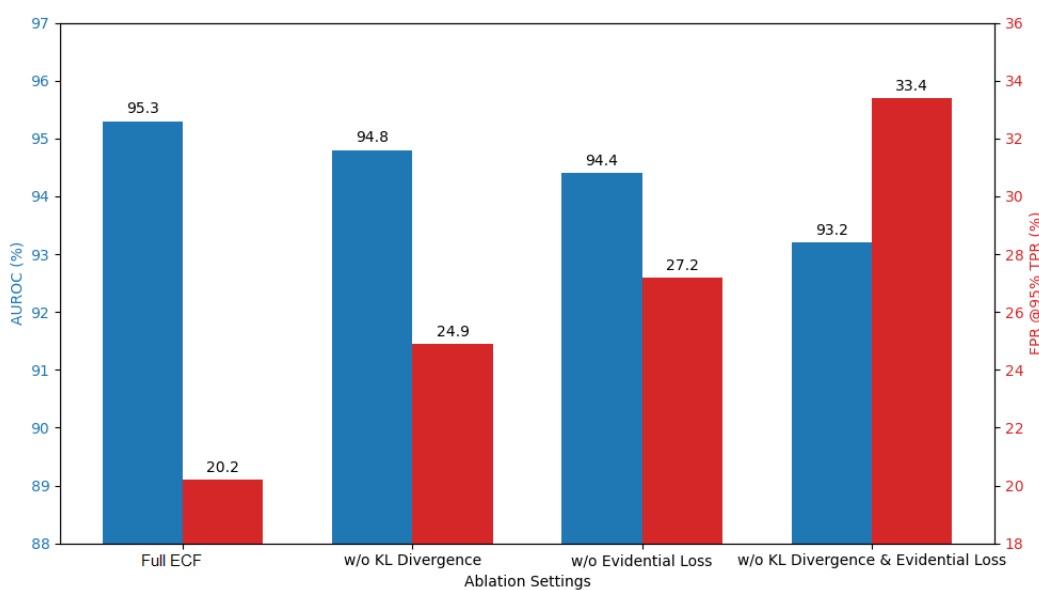

Figure 6: Ablation Analysis of Uncertainty Components in ECF for SVHN data

## B.5 STRENGTH OF ECF:

Table 8 highlights the uniqueness of the proposed Evidential Contrapositive Framework (ECF) against several state-of-the-art zero-shot OOD detection methods across a range of key attributes.

Table 8: Comparison of ECF with existing zero-shot OOD detection methods across multiple key attributes.

| Method | Zero-Shot Capability | Uncertainty Estimation | Negative Labels | Prompt Augmentation | Contrapositive Reasoning | Evidential Modeling | OOD-Aware Training |
|---|---|---|---|---|---|---|---|
| ECF (Ours) | ✓ | ✓(Epistemic + Aleatoric) | ✓ | ✓ | ✓ | ✓ | ✓ |
| CLIP-ZS | ✓ | | | | | | |
| MSP-CLIP | ✓ | ✓(Max Softmax) | | | | | |
| Energy-CLIP | ✓ | ✓(Energy Score) | | | | | |
| G-ODIN-CLIP | ✓ | ✓(Grad. Perturbation) | | | | | ✓(Partial) |
| ZOC | ✓ | | | ✓ | | | |
| CLIPN | ✓ | ✓(Abstention Score) | ✓ | | | | ✓ |
| TAG | ✓ | | | ✓ | | | ✓(Partial) |

Following describes the ECF's uniqueness:

- First to combine contrapositive reasoning with evidential deep learning for zero-shot OOD.
- Utilizes explicit negative labels during training for exclusion-aware generalization.
- Models both epistemic and aleatoric uncertainty, enabling calibrated decisions under distributional shift.
- Incorporates textual prompt augmentation for better generalization across unseen classes.

## B.6 LIMITATIONS OF ECF:

While our proposed Evidential Contrapositive Framework (ECF) achieves superior performance in zero-shot OOD detection, following are the limitations of ECF:

- **Prompt Dependency:** ECF's performance is sensitive to the quality and structure of textual prompts. Poorly constructed prompts may misalign with visual features, reducing the effectiveness of contrastive and contrapositive supervision.
- **Negative Label Semantics:** The approach relies on explicit negative labels derived from semantic distances. However, close semantic proximity between some negative and in-distribution labels may cause confusion, leading to reduced exclusion precision.
- **Uncertainty Modeling Overhead:** The inclusion of evidential deep learning for capturing epistemic and aleatoric uncertainty introduces additional computational cost and complexity compared to simpler OOD detection methods.

