# OpenReview forum: "Learning by Exclusion: An Evidential Contrapositive Framework for Zero-Shot OOD Detection"
_ICLR.cc/2026/Conference — ICLR 2026 Conference Withdrawn Submission_

### Official Review · Reviewer_BChV · 2025-10-27

**Soundness:** 2
**Presentation:** 3
**Contribution:** 2
**Rating:** 4
**Confidence:** 4

**Summary:**

### Method
- The authors point out two limitations in prior works:
    - Semantic ambiguity: CLIP-style models conflate semantically similar ID and OOD samples.
    - Overconfidence: Deterministic similarity scoring does not reflect predictive uncertainty.

- They introduce the Evidential Contrapositive Framework (ECF), which consists of three core components:
    - Contrapositive learning module.
    - Uncertainty modeling via evidential deep learning.
    - Zero-shot inference based on a dual-threshold strategy.

### Results
- ECF significantly outperforms prior zero-shot OOD methods, both in detection accuracy and in uncertainty calibration.

**Strengths:**

- ECF can model predictive uncertainty.
- ECF significantly outperforms prior zero-shot OOD methods.

**Weaknesses:**

### Major
- ECF requires training the model with five loss terms.
    - First, the baseline methods in the study are usually training-free, such as MCM, NegLabel, and AdaNeg. It is not very fair to compare ECF with these methods. Thus, I'm no sure if the proposed method really shows advantages.
    - Second, the training process needs five different loss terms with four weighting hyperparameters. Although the authors can tune the hyperparameters on a validation set, this framework looks too complex, especially compared with prior OOD methods.
    - Last, will the original abilities of CLIP be affected after training?

- Dual-threshold strategy
    - How do the authors decide the two hyperparameters, $\delta$ and $\gamma$ in the rejection strategy?
    - How to calculate AUROC when two parameters can affect the rejection decision?

### Minor
- The format of some citations is not correct. Please try to distinguish between \citep and \citet. For example, lines 119-120, table 1 and table 2.
- There are some incorrect uses of quotation marks, such as Line 185, Line 238, and Line 249.
- The formatting of the table could be improved. The first row appears very sparse with a lot of empty space, while the rows below look crowded.

**Questions:**

Please see the Weaknesses section.

---

### Official Review · Reviewer_spXn · 2025-10-29

**Soundness:** 2
**Presentation:** 1
**Contribution:** 2
**Rating:** 2
**Confidence:** 5

**Summary:**

The paper proposes the Evidential Contrapositive Framework (ECF) for zero-shot OOD detection with vision–language models. The key ideas are: (i) contrapositive/exclusion-based training that pushes image embeddings away from negative prompts; (ii) two auxiliary regularizers—Negative Prompt Alignment Loss (NPAL) and Contrapositive Similarity Regularization (CSR)—to further suppress unintended alignment with non-target classes; and (iii) evidential uncertainty modeling with a Dirichlet head to estimate both aleatoric and epistemic uncertainty for abstention and thresholding at inference.
The method is evaluated on CIFAR-10/100 and ImageNet-1K as ID datasets, with multiple OOD benchmarks. Reported metrics (AUROC, FPR@95) indicate improvements over several recent baselines, including CLIPN, NegLabel, AdaNeg, etc. The paper also includes ablations (contrapositive vs. evidential components) and uncertainty analyses illustrating separation between ID and OOD uncertainty distributions.

**Strengths:**

1. Conceptual integration: The combination of contrapositive-style negative prompt training with evidential (Dirichlet) uncertainty is novel and well-motivated for OOD rejection.
2. Practicality: The approach keeps the CLIP backbone frozen and adds a lightweight evidential head and losses. Claimed compute/memory overhead is modest, which is attractive for scaling.
3. Empirical performance: The method shows strong improvements across several benchmarks and provides comprehensive metrics (AUROC/FPR95) as well as ablations that isolate the contributions of contrapositive training and evidential modeling.

**Weaknesses:**

1. Conceptual clarity and problem definition (Zero-shot vs. training exposure).
Line 110 defines zero-shot OOD detection as “without requiring exposure to OOD examples during training.” This is incomplete and misleads the reader about the paper’s setting. In the zero-shot OOD literature, “zero-shot” usually indicates no exposure to task-specific labeled ID examples either (i.e., zero-shot classification on the ID label space via prompts only), or at minimum no additional training on ID images. In this paper, the method explicitly trains on ID images and ID prompts (“Models are trained using only the ID dataset classes,” Line 407). Therefore, the problem setting is more accurately  “N-shot OOD detection,” not “zero-shot OOD detection.” This affects fairness of comparisons and positioning:
The correct baselines should include few-shot or fully supervised ID training methods, depending on how many ID images are used, rather than methods that perform zero-shot inference without any ID training.
The paper should revise the terminology across the abstract, intro, and experiments to avoid conflating zero-shot with “no OOD exposure.” As written, it overstates novelty under a zero-shot claim.
2. Related work inaccuracies (CLIPN characterization).
Line 249 states: “CLIPN … introduces a learned ‘none-of-the-above’ token as a global OOD prototype.” This characterization is inaccurate or at least incomplete relative to CLIPN’s methodology, which also uses class-wise “not” tokens analogous to the negative prompt strategy proposed here. The paper should correct this description and more carefully contrast ECF with CLIPN along the actual axes of difference (e.g., evidential Dirichlet modeling, specific contrapositive loss structure, CSR/NPAL regularizers), rather than implying CLIPN only uses a single global “none-of-the-above” anchor.
3. On training data usage.
Line 407: “Models are trained using only the ID dataset classes.” This sentence understates the extent of training and may be read as minimal exposure, but the method appears to make extensive use of ID images during training to optimize the contrapositive and evidential objectives. If the approach indeed trains on all ID images, this should be clearly stated, and the comparison framed accordingly. If it uses a subset, the paper should report the exact number/percentage of ID images, selection protocol, and sensitivity to the amount of ID training data. Otherwise, the current phrasing risks overclaiming a zero-shot regime while benefiting from supervised ID training.

**Questions:**

See weaknesses.

---

### Official Review · Reviewer_SV1J · 2025-10-31

**Soundness:** 2
**Presentation:** 2
**Contribution:** 2
**Rating:** 4
**Confidence:** 5

**Summary:**

This paper introduces the Evidential Contrapositive Framework (ECF), a novel method for zero-shot out-of-distribution (OOD) detection. The core contribution is the integration of contrapositive reasoning from logic with evidential deep learning for uncertainty modeling. ECF trains a model to not only recognize in-distribution (ID) classes but also to explicitly learn what these classes are not by enforcing dissimilarity between image features and a set of negative text prompts. This exclusion-based learning is coupled with a mechanism that models class probabilities as a Dirichlet distribution, enabling the framework to disentangle and quantify both aleatoric (data) and epistemic (model) uncertainty. This dual approach allows ECF to reject an input if it has low similarity to all known classes or if the model exhibits high epistemic uncertainty. The authors conduct extensive experiments on several large-scale benchmarks, including ImageNet-1K, demonstrating that ECF significantly outperforms current state-of-the-art methods in terms of AUROC and FPR metrics.

**Strengths:**

1. The ability to model and disentangle epistemic and aleatoric uncertainty is a significant strength. This is particularly valuable in zero-shot settings, as it allows the model to make more reliable and interpretable decisions, distinguishing between inherent data ambiguity and a genuine lack of model knowledge.

2.  The paper is supported by comprehensive experiments across multiple challenging benchmarks (CIFAR-10, CIFAR-100, ImageNet-1K) and a variety of OOD datasets. The results consistently show that ECF achieves state-of-the-art performance, often by a significant margin, which strongly validates the effectiveness of the proposed framework.

3. The paper is well-written and clearly structured. The motivation is well-argued, the methodology is described in detail, and figures such as the t-SNE visualization (Fig. 1) provide an intuitive understanding of the method's core mechanism.

**Weaknesses:**

1. The paper's entire contribution rests on a training paradigm that appears fundamentally fragile and is not sufficiently documented to be verifiable. The method proposes a complex combination of multiple contrastive loss terms (contra, NAPL, CSR) but attempts to optimize them by only fine-tuning a few final layers on top of frozen CLIP encoders. Reshaping a high-dimensional and highly structured joint embedding space with such minimal trainable parameters is notoriously difficult and prone to catastrophic training collapse. The authors provide alarmingly little detail about the training procedure, failing to include learning curves, gradient norm analysis, or other indicators of a stable process. Most damningly, the provided code repository omits the crucial training-phase scripts, making an independent reproduction of the results impossible. This omission, combined with the lack of descriptive detail, severely undermines the paper's contribution and makes its central claims unverifiable.

2.  The framework's objective function, which explicitly supervises the distance to every negative label, is inherently unstable by design. In this setup, hard negatives—which are common and critical for effective OOD detection—will generate disproportionately large loss signals and gradients. This is highly likely to cause erratic training behavior and prevent stable convergence, especially given the limited capacity of the trainable head. The authors completely fail to acknowledge or address this significant stability risk. Without a detailed training dynamics analysis (e.g., examining the loss contribution from different negative samples), there is no evidence to suggest that the model can be reliably trained without extreme, unreported sensitivity to initialization or data ordering.

3. The paper's central claim rests on its ability to structure the embedding space for OOD rejection, yet it provides almost no empirical evidence to support this claim beyond a simplistic 2D t-SNE plot. A rigorous analysis would require visualizing the high-dimensional distributions of image features against their positive and corresponding negative text prompts, both before and after training, to demonstrate that the exclusion principle is actually being learned. Without this analysis, it is impossible to validate whether the complex loss functions are performing their intended roles or if the observed performance gains are merely an artifact of the training setup. The mechanism remains an unproven black box.

4. The empirical evaluation is built on a foundationally unfair comparison that artificially inflates the perceived performance of ECF. The proposed method, which requires a dedicated training stage on ID data, is predominantly benchmarked against training-free methods such as MCM, NegLabel, and AdaNeg. These methods operate under a much stricter and more challenging paradigm, often requiring no training data whatsoever. Comparing a fine-tuned model to zero-shot, training-free baselines is a critical methodological flaw. A valid evaluation would necessitate benchmarking against other state-of-the-art methods that also leverage a training or fine-tuning stage for OOD detection.

5. The final objective is a weighted sum of five different loss terms, controlled by four hyperparameters (λ1, λ2, λ3, λ4). Such a complex objective function is often brittle and extremely sensitive to the precise weighting of its components. The paper provides only a brief sensitivity analysis on a single dataset, which is insufficient to demonstrate robustness. It is highly probable that these weights require extensive and careful tuning for each specific ID-OOD dataset pair, which would severely limit the method's practical applicability and generality.

**Questions:**

None

---

### Official Review · Reviewer_r5Pp · 2025-11-01

**Soundness:** 2
**Presentation:** 3
**Contribution:** 2
**Rating:** 4
**Confidence:** 4

**Summary:**

This paper introduces a rejection-aware learning framework for OOD detection. The main motivation is that existing techniques rely on hard threshold to decide OOD samples, posing potential over-confidence issues. This paper thus introduce uncertainty modelling to OOD detection, leading to the so called rejection aware OOD detection method. Particularly, a prompt embedding based contrastive learning loss is presented as the contrapositive loss (Eq. 1), enforcing dissimilarity of the current class to all other class prompts. Furthermore, a negative prompt alignment loss is presented to further push away semantically similar classes, and a contrapositive similarity regularisation is to regularise the embedding space used in the contrapositive loss. Finally, both data uncertainty and model uncertainty are modelled as the predictive uncertainty to mitigate the over-confidence issue of OOD detection.

**Strengths:**

1. Targeting the over-confidence issue of OOD is meaningful.

2. The paper writing is clear, and the method is easy to be reimplemented.

3. The results are good compared with the listed comparison solutions.

**Weaknesses:**

1. The proposed contrapositive learning takes the relationships between classes and prompts into consideration for learning effective feature space. Similar idea has been explored in Ref1.

2. Evidential deep learning for OOD detection has also been explored in Ref 2 and Ref 3

3. With the same motivation, confidence aware OOD detection has also been explored in Ref 4.


Ref 1: SimLabel: Consistency-Guided OOD Detection with Pretrained Vision-Language Models. Arxiv 2025

Ref 2: Hyper-opinion Evidential Deep Learning for Out-of-Distribution Detection. NeurIPS 2024

Ref 3: Multi-label out-of-distribution detection via evidential learning. ECCV 2024 Workshop

Ref 4: CLIPScope: Enhancing Zero-Shot OOD Detection with Bayesian Scoring. WACV 2025

**Questions:**

The main problem of this paper is the lacking of sufficient background investigation. With the same motivation, zero-shot OOD has been equipped with confidence aware learning, either using Bayesian learning or evidential learning. In this case, the main question for this paper is to explain the innovation of the proposed solution with those existing techniques.

---

### Note · Authors · 2025-11-17

I have read and agree with the venue's withdrawal policy on behalf of myself and my co-authors.